# Refined Analysis of Chronic White Matter Changes after Traumatic Brain Injury and Repeated Sports-Related Concussions: Of Use in Targeted Rehabilitative Approaches?

**DOI:** 10.3390/jcm11020358

**Published:** 2022-01-12

**Authors:** Francesco Latini, Markus Fahlström, Fredrik Vedung, Staffan Stensson, Elna-Marie Larsson, Mark Lubberink, Yelverton Tegner, Sven Haller, Jakob Johansson, Anders Wall, Gunnar Antoni, Niklas Marklund

**Affiliations:** 1Section of Neurosurgery, Department of Medical Sciences, Uppsala University, 75185 Uppsala, Sweden; fredrik.vedung@neuro.uu.se (F.V.); niklas.marklund@neuro.uu.se (N.M.); 2Section of Radiology, Department of Surgical Sciences, Uppsala University, 75185 Uppsala, Sweden; markus.fahlstrom@radiol.uu.se (M.F.); elnamarielarsson@me.com (E.-M.L.); sven.haller@surgsci.uu.se (S.H.); 3Rehabilitation and Pain Centre, Uppsala University Hospital, 75185 Uppsala, Sweden; staffan.stenson@akademiska.se; 4PET Centre, Uppsala University Hospital, 75185 Uppsala, Sweden; mark.lubberink@radiol.uu.se (M.L.); anders.wall@akademiska.se (A.W.); 5Medical Physics, Uppsala University Hospital, 75185 Uppsala, Sweden; 6Division of Health, Medicine and Rehabilitation, Department of Health, Education and Technology, Luleå University of Technology, 97187 Luleå, Sweden; yelverton@tegner.com; 7Affidea CDRC Centre de Diagnostic Radiologique de Carouge SA, Clos de la Fonderie, 1227 Geneva, Switzerland; 8Section of Anesthesiology, Department of Surgical Sciences, Uppsala University, 75185 Uppsala, Sweden; jakob.johansson@surgsci.uu.se; 9Section of Nuclear Medicine and PET, Department of Surgical Sciences, Uppsala University, 75185 Uppsala, Sweden; 10Department of Medicinal Chemistry, Uppsala University, 75185 Uppsala, Sweden; gunnar.antoni@ilk.uu.se; 11Section of Neurosurgery, Department of Clinical Sciences Lund, Skåne University Hospital, Lund University, 22184 Lund, Sweden

**Keywords:** traumatic brain injury, sport related concussion, memory impairment, diffusion tensor imaging, white matter lesions, rehabilitative approaches

## Abstract

Traumatic brain injury (TBI) or repeated sport-related concussions (rSRC) may lead to long-term memory impairment. Diffusion tensor imaging (DTI) is helpful to reveal global white matter damage but may underestimate focal abnormalities. We investigated the distribution of post-injury regional white matter changes after TBI and rSRC. Six patients with moderate/severe TBI, and 12 athletes with rSRC were included ≥6 months post-injury, and 10 (age-matched) healthy controls (HC) were analyzed. The Repeatable Battery for the Assessment of Neuropsychological Status was performed at the time of DTI. Major white matter pathways were tracked using q-space diffeomorphic reconstruction and analyzed for global and regional changes with a controlled false discovery rate. TBI patients displayed multiple classic white matter injuries compared with HC (*p* < 0.01). At the regional white matter analysis, the left frontal aslant tract, anterior thalamic radiation, and the genu of the corpus callosum displayed focal changes in both groups compared with HC but with different trends. Both TBI and rSRC displayed worse memory performance compared with HC (*p* < 0.05). While global analysis of DTI-based parameters did not reveal common abnormalities in TBI and rSRC, abnormalities to the fronto-thalamic network were observed in both groups using regional analysis of the white matter pathways. These results may be valuable to tailor individualized rehabilitative approaches for post-injury cognitive impairment in both TBI and rSRC patients.

## 1. Introduction

Traumatic brain injury (TBI) affects more than 27 million people worldwide every year, often resulting in cognitive and functional deficits, the impairment of daily life functioning, and reduced quality of life [1,2,3,4,5].

Sport-related concussion (SRC), defined as a mild TBI, affects millions of athletes each year [6,7]. While most SRC-induced symptoms resolve within 2–3 weeks, headache, dizziness, confusion, and nausea may be long-lasting. Athletes with persisting symptoms beyond the first three months post-injury are often called the “miserable minority” reflecting a discrepancy into the debate concerning the psychological vs. organic origins of symptoms [8]. In fact, the treatment of these persisting cognitive deficits is challenging, in part since gross structural abnormalities on routine neuroimaging [e.g., structural magnetic resonance imaging (MRI), and computed tomography (CT)] are rare in repeated SRC (rSRC) compared with moderate and severe TBI [2,3,5,8]. Some of these athletes without traumatic damages visible at the routine morphological MRI sequences are not systematically considered for rehabilitation programs, and the possible long-term consequences of the brain injury are often neglected without a structural visible injury [9,10]. For these athletes, a diagnosis of a mild or major neurocognitive disorder caused by brain injury is commonly used [11], mostly based on the impaired cognitive function only [11,12]. Memory impairment has also often been linked to microstructural damage in the brain, and it affects patients both acutely and chronically after TBI [2,8,13,14].

The use of advanced MRI techniques, such as diffusion tensor imaging (DTI), can, in higher detail, reveal the presence of white matter injury that may cause injury-induced symptoms following TBI and SRC [8,15,16,17]. DTI can both qualitatively and quantitatively demonstrate pathology not detected by other modalities and is, therefore, an important tool not only in the research setting but in the clinical setting as well [18]. Altered DTI-based parameters in the subacute post-injury stage suggests different levels of white matter damage [19,20]. Robust evidence has shown a vulnerability of white matter bundles near the midline, such as the fornix and cingulum, to TBI-induced shearing forces [21,22,23]. Investigations of DTI-based parameters for specific regions such as the corpus callosum (CC), internal capsule (IC), and corona radiata (CR), indicates that baseline fractional anisotropy (FA) and mean diffusivity (MD) were associated with executive function and reaction time, respectively [24,25,26]. Other cognitive domains, including memory, may also depend on regional white matter integrity rather than generalized white matter injury [2,4,8,13,18,27,28,29,30]. Since there may be different subtypes of SRC, requiring specific rehabilitative therapies, refined white matter analysis is needed to understand the anatomical basis of the persisting symptoms [31].

Inter-subject differences in the mechanism of injury, as well as other biomechanical factors such as head and body composition, make it highly probable that, despite some commonalities, many areas of injury will differ among both patients and athletes [18,32]. The general interpretation of DTI-based-parameter alteration may therefore suffer these differences at group level, especially if the mean values of FA or MD are analyzed [18]. The further application of individualized assessments of regional brain injury are needed to realize the full potential of DTI as a research and clinical tool. These reasons may explain why, despite the general evidence, DTI is still not integrated in the clinical care of patients with TBI or SRC [9].

This work has two aims: first, to assess whether fiber tract analysis (local/regional) reveals alterations in DTI-based parameters not seen on whole tract analysis (global) nor morphological MRI in patients with TBI or rSRC athletes; second, to assess possible connections between local white matter alterations and long-term cognitive status in both groups.

## 2. Materials and Methods

### 2.1. Study Cohorts

Six patients were enrolled after a moderate (defined as Glasgow coma scale (GCS) score 9–13, loss of consciousness ≥ 5 min, and/or focal neurologic deficits [33]) to severe (GCS score ≤ 8) TBI and treated at the neuro-critical care unit ≥6 months at the department of neuro-surgery, Uppsala University hospital. Athletes of both sexes with rSRC and ≥6 months duration of post-concussion symptoms, according to the 4th edition of the Diagnostic and Statistical Manual of Mental Disorders, were recruited [34,35]. Ten age-matched HCs with no previous TBI, neurological condition, or current or previous active participation in any contact sport were recruited as a control group. The Regional Research Ethics Committee in Uppsala granted permission for the study (Dnr 2015/012). Written informed consent was obtained from all included patients/athletes and HCs. All research was conducted in accordance with the ethical standards given in the Helsinki Declaration of 1975, as revised in 2008.

### 2.2. Image Acquisition and Data Processing

High-resolution 3D-T1-weighted- (T1w), 3D-T2 fluid attenuated inversion recovery (T2-FLAIR), and susceptibility-weighted angiography (SWAN) images were acquired for morphological evaluation.

DTI was acquired with a single-shot echo-planar imaging sequence using the following imaging parameters: repetition time = 14,384 ms, echo-time = 78.6 ms, voxel size = 2 × 2 × 2 mm^3^, 73 slices, b-value = 1000 s/mm^2^ with 32-directions on a 3.0 Tesla PET/MR-system (Signa PET/MR, GE Healthcare, Milwaukee, Waukesha, WI, USA). Motion and eddy current correction of acquired DTI data was performed in eddy, FSL (http://fsl.fmrib.ox.ac.uk/fsl/fslwiki; last accessed 11 October 2021) [36]. The diffusion data were reconstructed in MNI space using q-space diffeomorphic reconstruction (QSDR) in DSI studio with a diffusion sampling length ratio of 1.25. The output resolution was 2 mm. Briefly, QSDR is a white-matter-based nonlinear registration approach that directly reconstructs diffusion information in MNI space. As such, parametric images of FA, AD, and RD were calculated in MNI space. Detail information on QSDR can be found in Yeh et al. [37]. The QSDR function also provides a R^2^-value between subject and MNI diffusion data. A value greater than 0.6 suggests a good registration result. A value greater than 0.6 was reported for all subjects. Major projection, commissural, and association white matter pathways (37 tracts in total) have previously been reconstructed within the HCP-1021 template following the anatomical criteria used for the Brain-Grid DTT reference atlas [38], which were applied to each subject.

Along-tract mapping was performed in DSI Studio. All included white matter pathways were stretched to correspond to straight lines. FA, AD, and RD were sampled along these lines and regressed using a kernel density estimator with default regression bandwidth at 1.0. Each point of these lines corresponds to one coordinate in the tract file generating indices for a given pathway. These were arranged from start to end with a corresponding DTI-based parameter for each given index for each subject.

### 2.3. Neuropsychology

The repeatable battery for the assessment of neuropsychological status (RBANS) [39], an objective test to measure neuro-cognitive functions including attention, verbal functions, visuospatial, immediate, and delayed memory, was performed by a trained neuropsychologist at the time of MRI investigation. Here, RBANS was used to estimate the neuro-cognitive burden in TBI and rSRC.

### 2.4. Data and Statistical Analysis

#### 2.4.1. General White Matter Changes/Damages

The Shapiro–Wilks test was performed on the underlaying data for each analysis to test for normality; parametric or non-parametric statistical methods were chosen accordingly. A Kruskal–Wallis test with Dunn’s test to correct for multiple comparisons was used to test whole white matter pathway DTI-bases parameter differences between TBI, rSRC, and HCs for all white matter pathways, respectively. Furthermore, the median and interquartile range was calculated for TBI, rSRC, and HC and for all white matter pathways and DTI-based parameters, respectively. General changes were defined as a whole white matter pathway with significantly decreased total average FA and significantly increased total average AD or RD [19,20]. For all performed statistical analysis, derived *p*-values < 0.05 were considered significant. Statistical analysis and graphic plots were created using GraphPad Prism 9 (v 9.3.1., GraphPad Software, La Jolla, CA, USA).

#### 2.4.2. Focal White Matter Changes/Damages

Focal along-tract analysis was performed using multiple unpaired *t*-tests comparing each white matter pathway index in TBI vs. HC and rSRC vs. HC for all DTI-based parameters and white matter pathways, respectively. The false discovery rate (FDR) was controlled using the Benjamini–Hochberg procedure with Q = 10%. In this regard, the number of false positives is kept below 10% of the number of significant indices. We chose a group of ten sequential significant indices to indicate a significant difference compared to HCs.

#### 2.4.3. Neuropsychological Test

For descriptive analysis, the average and standard deviation (SD) of each subpart of the RBANS index was calculated. A group comparison on each subpart between TBI vs. HCs and rSRC vs. HCs were performed using an unpaired *t*-test for independent samples for comparison between groups.

## 3. Results

### 3.1. Participants

Six TBI patients (four males, two females, mean age 27 ± 7), 12 rSRC athletes (6 males, 6 females, mean age 26 ± 7) and 10 HCs (five males, five females, mean age 26 ± 5) were enrolled. No TBI patient had any known psychiatric or psychological disorder prior to the injury. Four TBI patients had cerebral contusions, and 2 had multiple cerebral microbleeds (CMB) suggesting diffuse axonal injury. Of the TBI patients, four had a good clinical recovery on the Glasgow outcome scale (GOS 5) and two a moderate disability (GOS 4). rSRC athletes had attained a median of 6 sports-related concussions (range 3–10). Concussion symptoms were assessed by the sport concussion assessment tool (SCAT)-3 [40], displaying high symptom severity. Non-specific white matter lesions were found in 2 rSRC athletes. Clinical and radiological data are summarized in Table 1.

### 3.2. General White Matter Changes/Injuries

Differences in DTI-based parameters were observed in 29 white matter pathways in TBI patients compared to HCs (see Table 2). No rSRC athlete displayed significantly increased AD or RD or significantly decreased FA for any of the included white matter pathways (Table 2). On the other hand, six white matter pathways displayed significant difference in DTI parameters in rSRC compared with HC. The six structures displayed a completely different trend in DTI parameters with a higher FA and lower AD and RD compared with HC (Table 2). The median and interquartile range for all white matter pathways and DTI-based parameters are presented in electronic Appendix A (ESM1).

### 3.3. Focal White Matter Changes/Injuries

Three white matter structures displayed regional differences in DTI-based parameters in both groups compared to HCs. The left frontal aslant tract (FAT) displayed lower FA and higher RD in the fronto-opercular region in TBI patients. The same pathway displayed higher FA and lower RD in the supplementary motor area region in rSRC patients. The right anterior thalamic radiation (ATR) displayed lower FA and higher AD and RD in TBI patients localized mostly at the thalamic level. The rSRC group displayed higher FA and lower AD and RD mostly at the thalamic level. The genu of the corpus callosum (CC) showed lower FA, higher AD and RD in the TBI group close to the midline. In the rSRC group, the focal analysis showed higher AD and lower RD close to the midline (Figure 1).

### 3.4. Neuropsychology

Both TBI patients and rSRC athletes were impaired on the RBANS global outcome analysis compared with HC (TBI: 75 ± 24; rSRC: 80 ± 17; HC: 105.5 ± 2; TBI to HC: *p* = 0.03 rSRC to HC: *p* = 0.006). RBANS Memory scores were lower in both TBI, and SRC when compared to HC (*p* = 0.048 and *p* = 0.04; respectively). The RBANS Verbal scores were lower in rSRC athletes compared to HC (*p* = 0.048) although not in TBI (*p* = 0.07). No significative difference was detected among the groups for the other RBANS functional domains (Figure 2).

## 4. Discussion

The most important finding in our study was that focal regional differences in white matter pathways were observed at the chronic stage post-injury in both TBI patients and rSRC athletes who had a similar impairment on memory testing in comparison to healthy, age-matched controls (HC). For the development of novel treatments, and for precision rehabilitation, an enhanced understanding of the structural basis of persistent symptoms is crucial [41].

White matter changes, demonstrated by differences in DTI-based parameters, have been previously identified in TBI patients [19,21,25,28,30]. Furthermore, differences in DTI-based parameters are also seen in patients with only minor cognitive impairment and in patients with normal conventional MRI, supporting the role of DTI in detecting subtle injuries missed by other modalities [8,20,25,30,42]. In our study, TBI patients displayed significant changes in DTI-based parameters compared with HC in 29 of the 37 analyzed white matter structures in the global white matter analysis. Using the same analysis interpreting the mean values for each single white matter in rSRC athletes, no white matter pathways displayed global axonal or myelin abnormalities compared with HC. However, a different trend in DTI parameters compared with HC was observed in six white matter pathways compared with HC. It is suggested that different injury mechanisms and levels of energy produced by trauma to cortical and subcortical structures may be explaining factors [15,26,28,30]. This variability leads to a lack of consensus on the interpretation of chronic DTI-based parameters after TBI or rSRC [16,21,23]. In fact, DTI has been demonstrated to be sensitive to a wide range of group differences, although no specific trends have been consistently identified [29,43]. For this reason, we performed post-hoc analyses to investigate specific and focal injuries to white matter pathways. A similar method was previously used to analyze regional white matter changes after radiation therapy in DTI-based parameters [44] and to investigate white matter anomalies in a patient with visual snow syndrome [45]. We found a different level of regional differences in the same white matter networks only partially revealed by the global analysis. Several white matter pathways displayed regionally decreased AD and RD in their midline segments in rSRC, in agreement with previous reports in chronic rSRC [16,46,47,48]. On the other hand, there are conflicting results showing increased AD and RD indicative of damage to axonal fibers or myelin, respectively [43,49]. Plausibly, focal and/or incomplete damage to myelin or axonal fibers may not affect the entire pathway in terms of DTI-based metrics at the chronic stage. Incomplete damage to both myelin and axons in rSRC may induce a myelin repair process with a change in the dominant cell type contributing to the signal, with axonal bundles being replaced by astrocytes and/or microglia [15]. Hence, significantly higher AD and RD and, potentially, lower FA in rSRC patients could be expected at the chronic stage [16,21,42,50]. Animal models showed that neuronal shrinkage can occur in the absence of cell death or perisomatic axotomy [16,51]. Decreased AD and RD post injury may be related to such shrinkage, leading to less surface area along axons for parallel diffusion [16,51].

These data suggest that the time course of physiological recovery extends longer than initially thought in rSRC athletes [17]. In addition, previous inconsistent results matching cognitive outcome and DTI may depend on the different DTI analysis methods used [22,23,24,43,51,52]. Changes in DTI-based parameters may be subtle and difficult to detect due to technical factors (such as the number of DTI directions, the algorithm used for the analysis, normalization process, the technique of tractography, the choice of global or focal indices’ analysis among others) [52,53,54,55]. When the structural injury is focal and/or partially repaired, analysis of regional DTI-based metrics changes should be investigated to identify anatomical and possible links to functional information/dysfunction.

Both TBI patients and rSRC athletes performed significantly worse in the memory domain. This result agrees with other studies demonstrating impaired neuropsychological outcome in TBI patients and rSRC athletes during the subacute/chronic stage [2,4,5,7,20,23,39,56]. In patients with evidence of structural and functional abnormalities after TBI, effective cognitive rehabilitation interventions initiated post TBI enhance the recovery process and minimize the functional disability [57]. In rSRC patients, the persistence of symptoms long beyond the generally accepted time frame for recovery may reflect the development of post-concussion syndrome (PCS) [58,59]. There is, however, no consensus regarding the clinical neuroradiological criteria for PCS and, increasingly, the term persistent post-concussive symptoms (PPCS) is used [60]. Despite persisting symptoms, many of these athletes have normal MRI investigations [59]. The possible evidence of white matter alterations may represent an important factor to consider to plan early education [61], cognitive behavioral therapy [62], and/or aerobic exercise therapy [63], which have been shown to be effective in certain patients with post-concussion syndrome [59]. Moreover, since memory seems the common most affected domain in our two populations, the specific use of external memory aids and computer-assisted strategies may be indicated, since they have also been shown to improve attention, memory, and executive skills after TBI and therefore may be of help in SRC patients with memory impairment [10,64,65,66].

The indication for tailored cognitive rehabilitation based on the affected white matter networks and functional impairment would therefore be necessary to either restore or compensate for memory deficits or other functional impairments commonly debilitating to rSRC patients, as well as for TBI patients [67,68].

### 4.1. Functional Correlates of Regional White Matter Injuries

We found FAT, ATR, and CC to be among the abnormal white matter pathways due to significant variations in DTI-based parameters, at the regional analysis in both TBI and rSRC athletes. The two groups displayed similar functional outcome and a similar mid-line location for the DTI-based parameters changes at the regional analyses for the ATR and CC. FAT is a key component of a cortico-basal ganglia- thalamic-cerebellar circuit involved in action control [69,70]. In both hemispheres, the FAT plays a role in selecting among competing representations for actions that require the same motor resources (mainly the articulatory apparatus on the left hemisphere and the oculomotor and manual/limb action systems on the right hemisphere) [69,71,72]. Its damage has been related to impairment in speech and language functions, as well as executive functions, visual–motor activities, inhibitory control, working memory, and attention [69,71]. The damage/changes to segments of ATR can also be observed in the midline fluid percussion brain injury, a rodent model of diffuse TBI wherein memory deficits are observed [73]. Electrophysiological evidence suggests a crucial role for ATR connectivity in human memory formation, connecting the anterior and midline thalamic nuclear groups to the frontal lobe [74,75,76,77,78,79]. Commissural white matter pathways such as the genu and splenium of the CC are more vulnerable in their segment close to the midline due to a close relationship with fibrotic structures such as the falx cerebri [28,80,81]. Lesions or regional changes of the CC are known to disable the interhemispheric communication of multiple memory systems and disturb memory function [82,83,84], in particular in long-term verbal memory performance [85,86].

Taken together, these findings suggest that regional differences in the frontal and fronto-thalamic networks (FAT, ATR) may be key contributors to the poor memory performance observed in our study. The additional involvement of the genu of CC, may have resulted in a lack of interhemispheric modulation and resilience leading to a less compensable functional impairment [82,83,84,85,86,87]. Knowledge of the presence of such white matter changes is of importance to individualize treatment strategies [41].

### 4.2. Limitations

Our study has some limitations. The sample size of the three groups is small, especially the TBI group. Although the groups are thoroughly characterized and matched (age and gender) the injury mechanisms resulting in the TBI or rSRC were heterogeneous. Hence, our results should, therefore, be interpreted with caution. The limited sample size influenced our analysis and some of the variables such as age, gender, and hospitalization time were not included as confounding factors in this article. Since the TBI group is too small and other studies from our group did not identify significant differences among subgroups of rSRC athletes [88], we did not include the other variables in our analysis, which was beyond the aim of this article. We aimed to investigate differences in white matter changes with two different methods and to match the neuropsychological outcome with diffusion changes in the two groups. Further studies with larger cohorts are needed to clarify the role of the confounding factors in TBI and rSRC subjects at the chronic phase. We also believe that only longitudinal studies with repeated investigations would minimize the risk of hidden differences between the groups displaying trends for possible recovery. Another limitation is that we cannot exclude that possible comorbidities, such as post-traumatic stress disorder (PTSD), which has been described in TBI patients, might have contributed to the worse performance on memory domain by the TBI group [89]. On the other hand, the similar performance in rSRC athlete, the different background of the TBI patients included in this study and the absence of similar white matter alterations as previously reported may indicate a different reason for the results presented in this article.

To our knowledge, this is the first study applying focal injury analysis to white matter pathways in both TBI patients and rSRC athletes. Hence, another limitation may be linked to our methods with the creation of several small indices for each DTI-based parameter together with the intrinsic sensitivity to noise. We defined regional differences as a minimum of 10 consecutive indices reaching the pre-determined significant levels in comparison to controls at group level in both direction (increased or decreased when compared with healthy controls). We believe the FDR analysis may represent a reliable and solid way to discriminate the effects of confounds through the analysis of voxel’s spatial neighborhood. Another limitation is the possible role of cerebral contusions and CMB on our DTI results. Post-traumatic lesions such as contusions and CMB can be associated with lasting changes in perilesional white matter properties, even remotely from the lesion, and midsagittal CMB have been associated with cognitive decline after TBI [90,91,92]. Since we did not anatomically normalize CMB, an effect on the DTI results cannot be excluded. However, at the time of DTI, neither acquisition nor an expansive effect of the contusions was present, nor was a significant deformation of anatomical structures detected. The quality of the normalization process was carefully assessed before the analysis. In view of the differences in acquisition period and the different trend in AD and RD between TBI and rSRC, a major role for CMB is unlikely. Further studies with more advanced models of white matter investigations are necessary to minimize the effects of confounds and to, in detail, assess network vulnerability in TBI and rSRC.

## 5. Conclusions

In our study, TBI patients and rSRC athletes displayed different morphological images and different DTI abnormalities at the first level analysis of white matter but shared a similarly impaired memory performance at the chronic stage after injury. DTI analysis detected white matter pathway changes especially in TBI patients but underestimated focal or incomplete abnormalities when the white matter pathway was considered as one entity. At the regional analysis, similar regions of the left FAT, the genu of the CC, and the right ATR displayed different focal changes in both rSRC and TBI patients, reflecting possible differences in trauma and recovery mechanisms. The concomitant presence of white matter findings and the functional impairment observed in both TBI patients and rSRC athletes may suggest a long-term chronic impairment in some subgroups of patients despite the normal standard MRI images. This information seems crucial to better interpret the functional outcome of athletes with rSRC and to tailor individualized rehabilitative plans.

## Figures and Tables

**Figure 1 jcm-11-00358-f001:**
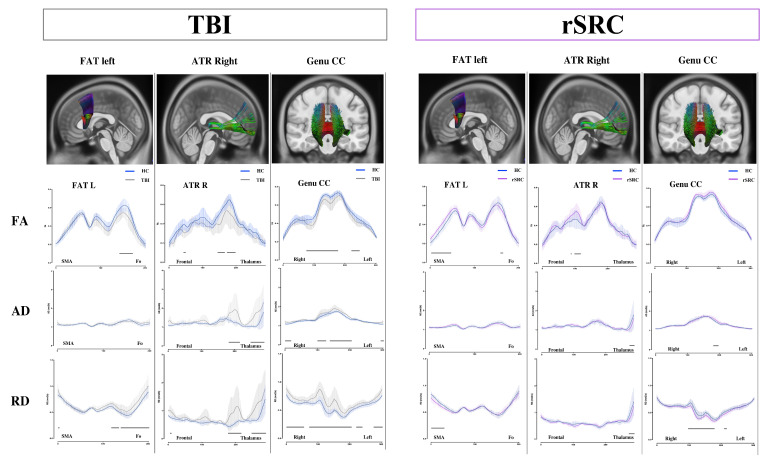
Focal analysis of three white matter pathways with white matter changes identified in both TBI patients and in rSRC patients compared with HC. False discovery rate (FDR) analysis was performed, as such significant differences are displayed as continuous lines on the x axis of each DTI-based parameter for each white matter pathway. The left frontal aslant tract (FAT) displayed lower FA and higher RD in the fronto-opercular region in TBI patients, while a higher FA and lower RD in the supplementary motor area region were found in the rSRC group. The right anterior thalamic radiation (ATR) displayed lower FA and higher AD and RD in the TBI group localized mostly at the thalamic level, while the rSRC group displayed higher FA and lower AD and RD mostly at the thalamic level. The genu of the corpus callosum (CC) showed lower FA and higher AD and RD in the TBI group close to the midline, while the rSRC group displayed a higher AD and lower RD close to the midline. TBI: traumatic brain injuries; rSRC: repeated sport related concussions; HC: healthy controls; AD: axial diffusivity; RD: radial diffusivity; FAT: frontal aslant tract; ATR: anterior thalamic radiation; CC: corpus callosum.

**Figure 2 jcm-11-00358-f002:**
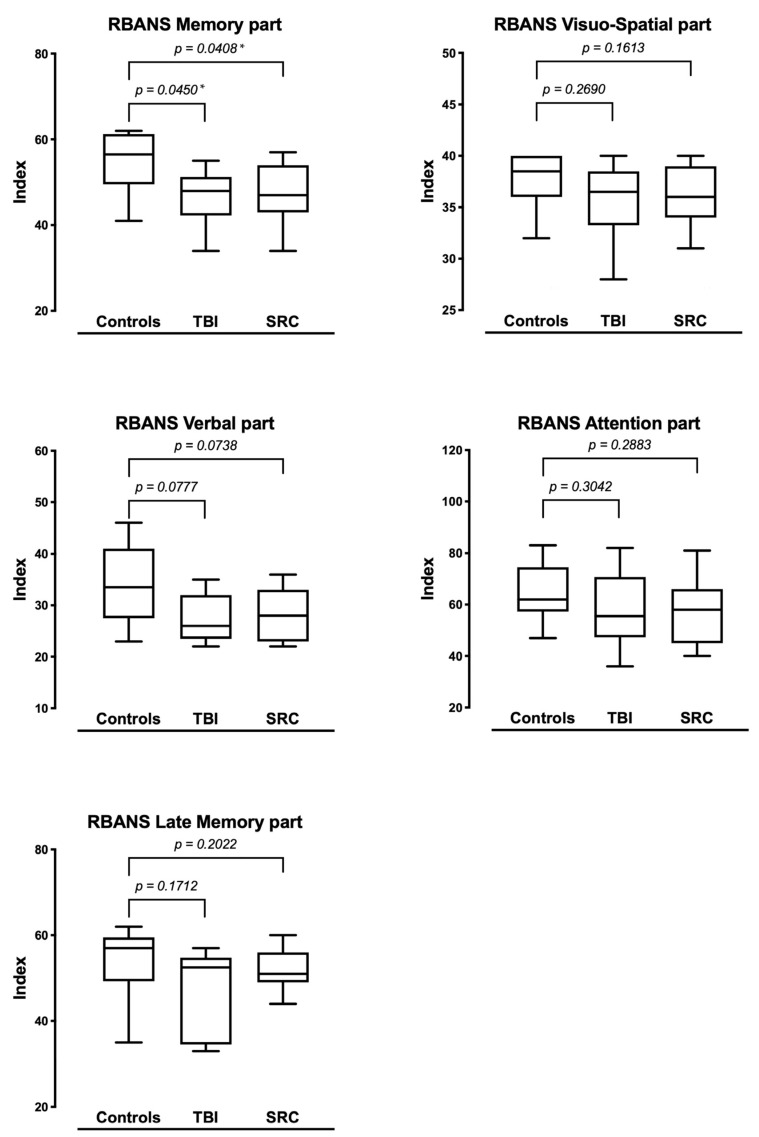
Each domain of RBANS is shown for TBI, rSRC, and HC. Significant differences in RBANS Memory score were detected between HC and TBI patients, as well as rSRC athletes. For the other domains of the RBANS, there were no significant differences among the groups. TBI: traumatic brain injuries; rSRC: repeated sport-related concussions; HC: healthy controls; RBANS: repeatable battery for the assessment of neuropsychological status. * Statistical significance with *p* value < 0.05.

**Table 1 jcm-11-00358-t001:** Summary of the clinical and radiological data of the three groups.

Clinical/Radiological Factors	Groups		
	TBI	SRC	HC
Number of patients	6	12	10
Age-mean (SD)	27 (7)	26 (7)	26 (5)
Gender-M/F	4/2	6/6	5/5
Concussions-no (range)	-	6 (3–10)	-
Contusions-no	4	-	-
DAI-no	2	-	-
Time since last TBI or SRC (months)	19 (8)	23 (6–132)	-
Length of Hospital stay (days)	17 (9)	-	-
Injury Mechanisms			
Fall	3	-	-
Motor vehicle accident	3	-	-
Sports-related	-	12	-
Neurologic status			
GCS at admission (range)	12 (5–14)	-	-
GCS at discharge (range)	14 (8–15)	-	-
GOS at the time of MRI (n of pts)	5 (4), 4(2)	-	-
Symptoms (SCAT)			
SSS (range)	-	48.5 (3–91)	-
NOS (range)	-	18 (2–22)	-

In athletes, concussion symptoms were assessed by the sport concussion assessment tool (SCAT). The symptom evaluation score lists 22 symptoms with a severity range of 0–6, and the symptom severity score (SSS) is the sum of all symptom scorings (range 0–132). The number of symptoms (NOS) is the sum of each symptom with a severity score between 1 and 6 (range 0–22). TBI: Traumatic brain injury, SRC: Sport-related concussions, and HI: healthy controls. DAI = diffuse axonal injury; GCS = Glasgow coma scale SCAT = sports concussion assessment tool; SSS = symptom severity score; NOS = number of symptoms; GOS: Glasgow outcome scale; non-parametric data (number of SRCs, time since last SRC, SSS, NOS, GCS) is presented as medians and range, and parametric data (age, time since TBI and length of hospital stay) is presented as means ± standard deviations (SD).

**Table 2 jcm-11-00358-t002:** Analysis of global white matter damage.

White Matter Structure	AD		FA		RD		Injured	
	TBI vs. HC	SRC vs. HC	TBI vs. HC	SRC vs. HC	TBI vs. HC	SRC vs. HC	TBI	SRC
AC	<0.0001	<0.0001	<0.0001	<0.0001	<0.0001	<0.0001	Y	N
AF L	<0.0001	<0.0001	<0.0001	<0.0001	0.4062	<0.0001	N	N
AF R	0.0006	>0.9999	0.0603	0.6801	0.0002	0.3323	Y	N
Internal capsule Anterior L	<0.0001	0.8410	<0.0001	0.0226	<0.0001	0.5070	Y	N
Internal capsule Anterior R	<0.0001	0.2387	0.0416	<0.0001	<0.0001	>0.9999	Y	N
FAT L	0.0007	>0.9999	0.0125	>0.9999	<0.0001	>0.9999	Y	N
FAT R	0.0178	>0.9999	0.0149	0.6267	0.0006	0.2962	Y	N
ATR L	<0.0001	0.1121	<0.0001	0.0588	<0.0001	0.5310	Y	N
ATR R	<0.0001	>0.9999	<0.0001	0.6583	<0.0001	0.2591	Y	N
Ci L	0.1709	0.0003	<0.0001	0.2943	<0.0001	0.0295	N	N
Ci R	<0.0001	<0.0001	0.0006	0.0026	<0.0001	0.0017	Y	N
CS L	<0.0001	0.1422	0.0006	>0.9999	<0.0001	>0.9999	Y	N
CS R	<0.0001	>0.9999	0.0322	0.8871	<0.0001	>0.9999	Y	N
External capsule L	<0.0001	0.8449	0.0374	0.0980	<0.0001	0.5370	Y	N
External capsule R	<0.0001	>0.9999	0.0410	>0.9999	<0.0001	0.6081	Y	N
FM	<0.0001	0.4771	0.0006	0.0028	<0.0001	<0.0001	Y	N
Fo L	<0.0001	<0.0001	<0.0001	0.0011	<0.0001	0.0001	Y	N
Fo R	<0.0001	<0.0001	<0.0001	<0.0001	<0.0001	<0.0001	Y	N
Genu CC	0.0019	>0.9999	<0.0001	0.3754	<0.0001	0.0060	Y	N
hSLF L	<0.0001	0.5733	0.0002	0.8185	<0.0001	0.8285	Y	N
hSLF R	<0.0001	>0.9999	0.0005	0.7111	<0.0001	>0.9999	Y	N
IFOF L	<0.0001	0.3086	0.0036	0.0009	<0.0001	0.1810	Y	N
IFOF R	<0.0001	>0.9999	0.2301	>0.9999	<0.0001	0.7752	Y	N
ILF L	0.0116	0.3316	0.0157	>0.9999	<0.0001	>0.9999	Y	N
ILF R	<0.0001	>0.9999	0.0140	>0.9999	<0.0001	0.6816	Y	N
MLF L	0.1890	0.0226	0.2841	0.0774	<0.0001	0.3796	N	N
MLF R	0.5220	>0.9999	0.0011	0.0105	<0.0001	<0.0001	N	N
OR L	<0.0001	0.6411	0.0207	0.5038	<0.0001	>0.9999	Y	N
OR R	<0.0001	>0.9999	>0.9999	>0.9999	<0.0001	0.8799	N	N
Internal capsule posterior L	<0.0001	0.0622	<0.0001	>0.9999	<0.0001	>0.9999	Y	N
Internal capsule posterior R	<0.0001	>0.9999	0.0144	0.8507	<0.0001	>0.9999	Y	N
UF L	<0.0001	0.3353	0.1033	0.0110	<0.0001	0.0007	Y	N
UF R	0.1262	0.4169	0.0729	0.7228	0.0022	0.0070	N	N
VO L	0.5318	>0.9999	0.0022	>0.9999	<0.0001	>0.9999	N	N
VO R	0.4149	0.1643	0.1207	0.0106	0.0240	0.3355	N	N
vSLF L	0.0027	0.0081	0.2355	>0.9999	0.0249	>0.9999	Y	N
vSLF R	<0.0001	0.0105	0.0012	0.5643	<0.0001	0.5110	Y	N

Adjusted *p*-values are presented for each comparison between traumatic brain injury (TBI), healthy controls (HC), and repeated sport-related concussions (rSRC) and HC including, all DTI-based metrics and white matter pathways. Values in red emphasize results wherein the axial diffusivity (AD) or radial diffusivity (RD) values are increased in comparison to HC, defining the criteria for white matter pathway injury. Values in green emphasize results wherein AD or RD decreases, which is contradictory to the theory presented. Y: yes (injured), N: not injured, L: left, R: right; AC: anterior commissure; AF: arcuate fasciculus; FAT: frontal aslant tract; ATR: anterior thalamic radiation; Ci: cingulum; CS: cortico-spinal tract; FM: forceps major; Fo: fornix; CC: corpus callosum; hSLF: horizontal component of superior longitudinal fasciculus; IFOF: inferior fronto-occipital fasciculus; ILF: inferior longitudinal fasciculus; MLF: middle longitudinal fasciculus; OR: optic radiation; UF: uncinate fasciculus; VO: vertical occipital fasciculus; vSLF: vertical component of superior longitudinal fasciculus.

## Data Availability

The data that support the findings of this study are available on request from the corresponding author. The data are not publicly available due to privacy and ethical restrictions.

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
