# Peer review of "Refined Analysis of Chronic White Matter Changes after Traumatic Brain Injury and Repeated Sports-Related Concussions: Of Use in Targeted Rehabilitative Approaches?"

_jcm, 2022, doi:10.3390/jcm11020358_

Round 1

Reviewer 1 Report

Summary:

Latini and colleagues examined the white-matter microstructural alterations in chronic TBI using in vivo imaging in chronic TBI and rSRC in comparison with the healthy controls. They examined the differences between chronic TBI, rSRC, and HC to define the distribution of post-injury 23 regional white matter changes after TBI and rSRC. Their results showed altered white matter microstructures, revealed using focal injury analysis, associated with the worse memory performance in rSRC and TBI in comparison to HC. These findings would suggest the need to tailor individualized rehabilitative approaches for cognitive impairment in both TBI and rSRC.

I find the study to be complementary to the literature, however, there are several concerns about the paper that would require attention from the authors.

Major concerns:

  1. The sample size is a major concern in the current study.
  2. Moderate-to-severe TBI patients are usually associated with the presence of lesions. Therefore, it is important to consider this while looking into the data. Also the presence of the lesion would affect the registration and normalization of the data to the MNI. In addition, it should be added as a factor in the analysis.
  3. The order of the pre-processing is problematic. To avoid any effect of the normalization to MNI on the DTI directions, studies usually estimate the DTI measures then perform the normalization not as mentioned in the manuscript. In addition, no information was provided on the method used for the normalization of the DTI data to MNI.
  4. In the statical analysis: 2.4.1, the authors mentioned using “ANOVA with Dunnett’s multiple comparisons test”, which assumes the normal distribution of the data. Was the check of data normality performed? If not, it should be done, and in case the data is not normally distributed, then the use of other methods such as Wilcoxon signed-rank or other methods of analysis would be required.
  5. No clear threshold of the p value for the statical significance.
  6. The duration post-TBI and the should be used as confounding factors in the analysis.
  7. In the statistical analysis, several confounding factors should be considered including age, gender, education, the number of reported TBI incidents, time since injury, hospitalization time. For example, previous studies suggested that females might suffer from more white-matter alterations as compared to males following TBI. (see Rubin TG, et al. MRI-defined white matter microstructural alteration associated with soccer heading is more extensive in women than men. Radiology. 2018;289:478–486.).
  8. The authors mentioned that they only included moderate-to-severe TBI, however in Table 1, it is shown that the TBI group had GCS at admission ranged 5-14. The definition of moderate is with GCS 9-12. This is a problem.
  9. In Table 2, the authors showed only the P-value, however it is helpful to have the mean and standard deviation of the FA, AD, and RD in each of the white-matter tracts in each group.
  10. In Table 2, it is mentioned that green emphasize results where AD or RD decreases, and red emphasize results where AD or RD increased. However, there are some regions with p<0.05 which has no colour code. Please revise Table2.
  11. In the discussion, the authors mentioned that there are several possible reasons for the difference in changes observed in the study compared to the literature. One of the reasons is the technical factors. Addressing the issues described before could help in clarifying this effect.  
  12. In the limitation, it is important to note that the changes in the TBI group might be contributed also to other comorbidities including PTSD as shown in Mohamed et al. (Biol Psychiatry Cogn Neurosci Neuroimaging 2021).

Minor issues.

  1. It is not reported which software package used for the statical analysis of the neuroimaging data.
  2. In line 165, “DAI=. GCS=” are wrong .
  3. In line 284, the authors wrote “significantly worse p”, this is unclear sentence, please revisit.
  4. In line 310, “Despite the correlation analysis with the RBANS memory score was negative,”, it implies negative correlation. Please rephrase.
  5. There are a few grammar and spelling mistakes, please revise.

Author Response

Response to reviewer #1

  1. The sample size is a major concern in the current study.

A: We are aware of the limitation of the study due to the small number of subjects as stated in the discussion and limitation section. However, our aim was to investigate different level of white matter abnormalities in different populations that are usually treated and rehabilitated in different ways. We hope that our initial results and methodological considerations may lead to further studies with larger cohorts of subjects to increase our knowledge on white matter lesions in the chronic stage after TBI and rSRC for a better and tailored approach after the hospitalization.

  1. Moderate-to-severe TBI patients are usually associated with the presence of lesions. Therefore, it is important to consider this while looking into the data. Also the presence of the lesion would affect the registration and normalization of the data to the MNI. In addition, it should be added as a factor in the analysis.

A: All the subjects were enrolled in a chronic stage after traumatic event. As stated in the text, four TBI patients had cerebral contusions, and 2 had multiple cerebral microbleeds (CMB) suggesting diffuse axonal injury. None of these lesions created expansive effect at the time when DTI was performed and the normalization process and the quality of the reconstructed images was assessed as described in the comment 3.

  1. The order of the pre-processing is problematic. To avoid any effect of the normalization to MNI on the DTI directions, studies usually estimate the DTI measures then perform the normalization not as mentioned in the manuscript. In addition, no information was provided on the method used for the normalization of the DTI data to MNI.

A: DTI data was normalized to MNI using the QSDR function in DSI Studio. QSDR is a white matter based nonlinear registration approach that directly reconstructs diffusion information in MNI space. We have added more details on this approach as well as the correct reference to the method. Furthermore, with this method we do not expect that lesions would present a significant issue. Built-in quality assurance (R2 value between native and MNI space) for each subject suggested good normalization. This was also confirmed by visual quality assessment.

  1. In the statical analysis: 2.4.1, the authors mentioned using “ANOVA with Dunnett’s multiple comparisons test”, which assumes the normal distribution of the data. Was the check of data normality performed? If not, it should be done, and in case the data is not normally distributed, then the use of other methods such as Wilcoxon signed-rank or other methods of analysis would be required.

A: We have revised the statistical section. Normality test was performed using Shapiro Wilks test on all underlaying data. For the global analysis the data is not normal distributed, as such, we re-calculated the statistics using Kruskal-Wallis Test with Dunn’s test to correct for multiple comparisons. Table 2 has been updated with new calculated p-values.  

  1. No clear threshold of the p value for the statical significance.

A: We have added a clear statement on the p value threshold

  1. The duration post-TBI and the should be used as confounding factors in the analysis.

A: Since all the subjects were investigated at least after 6 months after the last traumatic event, we considered the whole population as at the chronic phase and therefore, the time after the last event was not considered as a confounding factor.

  1. In the statistical analysis, several confounding factors should be considered including age, gender, education, the number of reported TBI incidents, time since injury, hospitalization time. For example, previous studies suggested that females might suffer from more white-matter alterations as compared to males following TBI. (see Rubin TG, et al. MRI-defined white matter microstructural alteration associated with soccer heading is more extensive in women than men. Radiology. 2018;289:478–486.).

A: This is of course a valid point. We deliberately chose not to adjust for confounding factors mainly because of the small sample size and uneven distribution of for example, gender in the TBI group.  Furthermore, Rubin et al did suggest that female might suffer from more white matter alterations, however, they found significant differences between gender in FA and AD, but not RD. Our focus are especially directed towards AD and RD, since FA comprise of the same eigenvectors which defines AD and RD.

  1. The authors mentioned that they only included moderate-to-severe TBI, however in Table 1, it is shown that the TBI group had GCS at admission ranged 5-14. The definition of moderate is with GCS 9-12. This is a problem.

A: As mentioned in the text, we included moderate to severe TBI patients. We revised the text to clarify the inclusion criteria. Six patients (4 males and 2 females; mean age 27 ± 7 years) with a moderate (defined as Glasgow Coma Scale (GCS) score 9–13, loss of consciousness ≥5 min and/or focal neurologic deficits (Ingebrigtsen et al., 2000) – to severe (GCS score ≤8) TBI and treated at the neuro- critical care unit ≥6 months previously at the department of Neuro- surgery, Uppsala University hospital

  1. In Table 2, the authors showed only the P-value, however it is helpful to have the mean and standard deviation of the FA, AD, and RD in each of the white-matter tracts in each group.

A: We added median and interquartile range for all DTI-based metrics and white matter tracts as a supplemental table. Cells filled with grey indicate statistical significance different compared to HC as described in the table caption.

  1. In Table 2, it is mentioned that green emphasize results where AD or RD decreases, and red emphasize results where AD or RD increased. However, there are some regions with p<0.05 which has no colour code. Please revise Table2.

A: the table is now revised with all the significant p-values colour-coded.

  1. In the discussion, the authors mentioned that there are several possible reasons for the difference in changes observed in the study compared to the literature. One of the reasons is the technical factors. Addressing the issues described before could help in clarifying this effect.  

A: As mentioned in the discussion both the software used and the techniques used for the DTI reconstructions may lead to variability of the results. We have now specified with some more examples possible technical factors affecting the results of global and focal DTI analyses.

  1. In the limitation, it is important to note that the changes in the TBI group might be contributed also to other comorbidities including PTSD as shown in Mohamed et al. (Biol Psychiatry Cogn Neurosci Neuroimaging 2021).

A: We have now added the potential contribution to white matter abnormalities due also to the PTSD as suggested. However, the Study of Mohamed et al., analysed subjects of older age with some familiar history of dementia and many of the subjects reported even trauma before the war. We believe that our results in the young athletes of the study may rather reflect a subacute stage without any other psychiatric diseases/conditions detected after the traumatic event. 

Minor issues.

  1. It is not reported which software package used for the statical analysis of the neuroimaging data.

A: All the information have now been provided in the methods section.

  1. In line 165, “DAI=. GCS=” are wrong .

A: the sentence is now correct

  1. In line 284, the authors wrote “significantly worse p”, this is unclear sentence, please revisit.

A: the sentence is now correct

  1. In line 310, “Despite the correlation analysis with the RBANS memory score was negative,”, it implies negative correlation. Please rephrase.

A: the sentence is now revised.

  1. There are a few grammar and spelling mistakes, please revise.

A: we have now revised spelling and grammar mistakes.

Reviewer 2 Report

The study is good in general. The Followings are the points to be corrected.

  1. In-Line # 27, what does DTI stand for?
  2. In-Line # 57, “[11]mostly” needs space.
  3. In-Line # 284, “worse p in the memory” what does p stand for?
  4. In-Line # 317, the word “in” has been written two times.
  5. Needs minor English editing.

Author Response

Response to reviewer #2

Comments and Suggestions for Authors

The study is good in general. The Followings are the points to be corrected.

  1. In-Line # 27, what does DTI stand for?

A: the sentence is now consistent with previous mention of DTI.

  1. In-Line # 57, “[11]mostly” needs space.

A: the sentence is now correct.

  1. In-Line # 284, “worse p in the memory” what does p stand for?

A: the sentence is now correct.

  1. In-Line # 317, the word “in” has been written two times.

A: the sentence is now correct.

  1. Needs minor English editing.

A: we have now revised the English editing.